# How Important are Network Weights? To What Extent do they need an Update?

## Abstract

In the context of optimization, a gradient of a neural network indicates the amount a specific weight should change with respect to the loss. Therefore, small gradients indicate a good value of the weight that requires no change and can be kept frozen during training. This paper provides an experimental study on the importance of a neural network weights, and to which extent do they need to be updated. We wish to show that starting from the third epoch, freezing weights which have no informative gradient and are less likely to be changed during training, results in a very slight drop in the overall accuracy (and in sometimes better). We experiment on the MNIST, CIFAR10 and Flickr8k datasets using several architectures (VGG19, ResNet-110 and DenseNet-121). On CIFAR10, we show that freezing 80% of the VGG19 network parameters from the third epoch onwards results in 0.24% drop in accuracy, while freezing 50% of Resnet-110 parameters results in 0.9% drop in accuracy and finally freezing 70% of Densnet-121 parameters results in 0.57% drop in accuracy. Furthermore, to experiemnt with real-life applications, we train an image captioning model with attention mechanism on the Flickr8k dataset using LSTM networks, freezing 60% of the parameters from the third epoch onwards, resulting in a better BLEU-4 score than the fully-trained model. Our source code can be found in the appendix.

## 1 Introduction

The immense success of deep neural networks we are witnessing since the deep learning revolution occurred is surprising. A large variety of vision and language applications ranging from image classification, object detection, image synthesis, image super-resolution, image captioning, language modeling....etc. has proved that neural networks possess a powerful capability of learning very complex data. However, training these networks to perform as expected is very time-consuming and requires powerful graphical processing units (GPUs). A recently published open-source project by NVIDIA[1] claimed that training a generative adversarial network (GAN) took more than 6 days on 8 Tesla V100 GPUs.

However, we argue that a lot of parameters involved during training are important for update only for the first few epochs (in our experiments, the first two epochs only), and can be frozen for the rest of the training epochs. The backpropagation algorithm is the base algorithm used to optimize deep neural networks. For each weight, a gradient is computed with respect to the loss which indicates the amount a weight should change. Large gradients correspond to a large change that will occur in the weight, while small ones (near to zero) indicate that the weight is nearly optimized and does not need much change. In particular, if a gradient for a particular weight is zero or close to zero, this means that it has either reached its optimal solution, or it is stuck at a saddle point. The former means that the weight has a good value and is less likely to change throughout the training and can be kept frozen. In this paper, we wish to show the redundancy of weights in a neural network that have no influence and can be kept frozen during training. In particular, we demonstrate that fully training a model with all its weights is required for the first two epochs only. To justify this, we propose an experimental technique named Partial Backpropagation, which freezes weights that have gradients very near to zero and are less likely to change, with the rest of the weights trained normally. This induces a very slight drop in accuracy (and no harm in accuracy for lesser freezing). An overview of

---

[1]https://github.com/NVlabs/stylegan

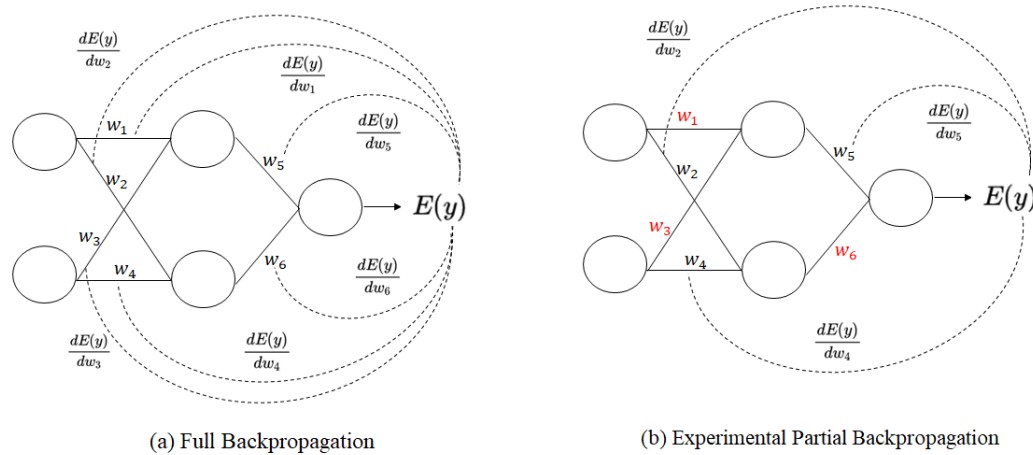

Figure 1: The difference between (a) full backpropagation and (b) our proposed experimental technique. In (b), weights which have nearly zero gradients are less likely to change throughout the training and are therefore kept frozen without being updated. These frozen weights are shown in red.

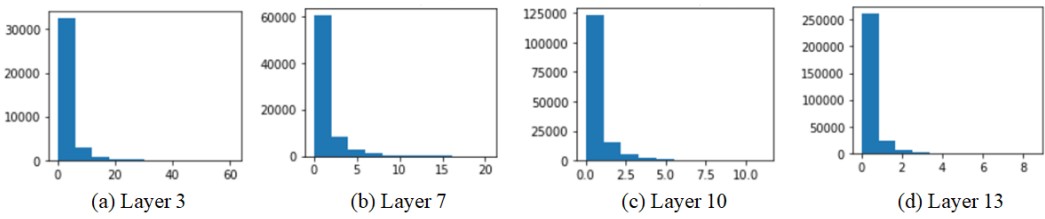

Figure 2: The histogram plot of gradients for (a) layer 3, (b) layer 7, (c) layer 10 and (d) layer 13 of a VGG19 convolutional network. A large number of gradients have values very close to 0, indicating that a lot of weights in these layers have reached to a good value and are less likely to change throughout the training.

our experimental technque is shown in Figure 1. Note that in Figure 1(b), the red weights are frozen and not removed or zeroed out.

We can further visualize the histogram of gradients across the network layers to have a better understanding of their distributions. In Figure 2, we visualize the distribution of gradients from several layers in a VGG19 convolutional network (Simonyan & Zisserman, 2015). In particular, we visualize the gradients of layers 3, 7, 10 and 13 after training for 2 epochs. We can see a large number of gradients with values very near to zero, suggesting that a lot of weights in these layers have already been optimized and are less likely to change throughout the training.

## 2 RELATED WORK

We discuss two topics closely (but not exactly) related to our work, namely weight pruning and progressive freezing of hidden layers.

### 2.1 WEIGHT PRUNING

The reduction of the heavy inference cost of deep networks in low-resource settings is the primary purpose of weight pruning. Neural Networks usually contain a large number of parameters that

make them hard to bring into effective action on embedded devices such as mobile phones. The hardship here happens due to both the network size and evaluation time. Significant redundant parameters could be reduced resulting in a compressed network with the unnecessary computation being alleviated while maintaining the same accuracy. In short, the main objective of weight pruning techniques is the reduction of the energy and storage needed to process inference on deep neural networks. Hence, they can be deployed on embedded devices. (Han et al., 2015) learn only the significant connections to reduce storage. A neural network is first trained to learn the significant connections. Subsequently, the insignificant connections are pruned out and the remaining weights are then fine-tuned. In particular, after the initial training phase, all weights lower than a certain threshold are eliminated. This results in a sparse network which is then re-trained so the remaining weights can compensate for the removed ones. The phases of pruning and retraining is an iterative process. After many iterations, the minimum number of weights could be found. In (Babaeizadeh et al., 2016), two neurons with highly correlated activations are merged into one instead of removing a neuron, which keeps the signals inside the network as close as possible to the original one. If the highest correlated neurons are not fully correlated, merging them into one neuron could possibly alter the accuracy factor. (Han et al., 2017) proposed a dense-sparse-dense training flow. Other works also focus on pruning convolutional filters for efficient inference. In (Molchanov et al., 2017), a new criterion based on Taylor expansion which approximates the change in the cost function induced by pruning network parameters is proposed. Similarly, (Li et al., 2017) proposed an acceleration method for convolutional networks where different filters which have a small influence on the output accuracy are pruned. By pruning whole filters in the network with their corresponding feature maps, the computation cost is greatly minimized. Moreover, (Liu et al., 2017) enforces channel-level sparsity in the network to reduce the model size of a CNN, which decreases the run-time memory and lower the number of computing operations with no harm on the accuracy factor. (Zhang et al., 2018) presented a systematic weight pruning framework of DNNs using the alternating direction method of multipliers (ADMM). A recenetly proposed netowrk, namely slimmable neural network (Yu et al., 2018) presents a simple and general method to train a single neural network which is capable of being exucted at different widths. Rather than training an individual network with different width configurations, a shared network with switchable batch normalization could be trained. During runtime, the network can change its width on the fly according to the resource limitation. Other works on compressing deep neural networks include (Luo et al., 2017) and (Ullrich et al., 2017).

However, an important aspect of weight pruning models is that deep neural networks are mainly trained for the objective of reducing computation at testing time for efficient implementation on hardware systems, which is an iterative training process and requires a lot of procedures and algorithms to be involved during training, which in turn increases the training time. In contrast to our work, we wish to demonstrate the number of redundant parameters, which if frozen and not zeroed out, can have a very slight (or no) influence on the overall accuracy. Since the two objectives are different, we cannot directly compare weight pruning methods with our experimental technique.

## 2.2 PROGRESSIVE FREEZING OF HIDDEN LAYERS

In (Brock et al., 2017), the authors proposed training each layer for a set portion of the training schedule, progressively "freezing out" layers and excluding them from the backward pass. This technique is motivated by the fact that in many deep networks, initial layers consume most of the cost, but have the fewest parameters and tend to converge to reasonable simple structures, implying that they do not require as much update as the later layers where most of the parameters are populated. This concept aims to reduce training time, however works on freezing complete layers which may contain informative gradients, and corresponding weights avoid getting updated. The study reports training time improvements with loss/no loss in accuracy, however does not report the number of parameters frozen in the overall network. Moreover, the study "progressively" freezes some layers in the network, implying that important weights at specific times in the training schedule experience an update. Therefore, it is also hard to directly compare Freezout with our experimental technique.

## 3 EXPERIMENTAL TECHNIQUE

As mentioned in Section 1, weights with gradients very near to zero imply that the particular weight is nearly optimized and is less likely to change throughout the training and can therefore be kept

frozen. Our experimental technique freezes weights that have gradients very near to zero from the third epoch onwards, while training rest of the weights normally. All weights with gradients below a particular threshold are frozen. In this section, we first describe how we calculate the threshold of gradients and then elaborate on the proposed experimental technique.

## 3.1 THRESHOLDING TECHNIQUE

We start by extracting the non-zero gradients of the neural network. After training, a slight amount of weights may posses zero gradients, which are not considered when performing thresholding. Given the gradients of each layer $l_i$ where $i$ indicates the layer number, we have: $G = \left(\frac{dE(y)}{dw}\right)^{l_1}, \left(\frac{dE(y)}{dw}\right)^{l_2}, \ldots\ldots, \left(\frac{dE(y)}{dw}\right)^{l_n}$. Note that each term $\left(\frac{dE(y)}{dw}\right)^{l_i}$ consists of all the gradients in the layer $i$. For convenience, we divide each gradient by the corresponding learning rate $\alpha$ to obtain the original gradient (non-scaled by the learning rate), and take its absolute value $\left|\frac{dE(y)}{dw}\right|$. This is because we are interested in the magnitude of the gradient, and not its direction. We then plot the kernel density estimation (KDE) of $G$, given by the equation:

$$\hat{f}(x) = \frac{1}{n}\sum_{i=1}^{n} K\left(\frac{x - x(i)}{h}\right) \tag{1}$$

where $K$ is the kernel function and $h$ is the kernel bandwidth. From $\hat{f}(x)$, we find the region which has a high likelihood of the gradients observation, obtaining the new range of gradients $G_n$. The new range $G_n$ is then divided into $b$ sub-ranges and the mean of each sub-range $b$ is obtained to form $G_d = \mu_1^g, \mu_2^g, \ldots\ldots, \mu_b^g$, where each gradient in $G_n$ is assigned to its corresponding mean value. We set $b$ to 300. We then apply Otsu thresholding technique to find the optimal threshold value. We re-write the mathematical formulation of otsu's method operating on the mean values in $G_d$. The within-class variance is calculated as:

$$\sigma_w^2(t) = q_1(t)\sigma_1^2(t) + q_2(t)\sigma_2^2(t) \tag{2}$$

where,

$$
\begin{array}{lll}
q_1(t) = \sum_{i=1}^{t} P(i) & \& & q_2(t) = \sum_{i=t+1}^{b} P(i) \\
\mu_1(t) = \sum_{i=1}^{t} \frac{iP(i)}{q_1(t)} & \& & \mu_2(t) = \sum_{i=t+1}^{b} \frac{iP(i)}{q_2(t)} \\
\sigma_1^2(t) = \sum_{i=1}^{t} [i - \mu_1(t)]^2 \frac{P(i)}{q_1(t)} & \& & \sigma_2^2(t) = \sum_{i=t+1}^{b} [i - \mu_2(t)]^2 \frac{P(i)}{q_2(t)}
\end{array} \tag{3}
$$

where $t$ is the mean value and represents the threshold at each of the 300 steps, (i.e. $t = \mu_1^g, \mu_2^g \ldots\ldots \mu_b^g$). After calculating the within-class variance of all values in $G_d$, we select the mean value from $G_d$ with the least within-class variance as our final threshold value.

## 3.2 EXPERIMENTAL ALGORITHM

We first start by training the network with full backpropagation for 2 epochs. It is necessary that the network first understands the data it is dealing with and performs weights update. After the second epoch, the network is considered to have known what weights are optimized and what weights further need optimization. We then perform full backpropagation on the first batch of the third epoch, however before updating the weights, we perform the gradient mask generation. Given the Jacobian matrix of the gradients with respect to the loss $F$ at the first batch of the third epoch:

$$
\mathbf{J} = \begin{bmatrix}
\frac{\partial F_1}{\partial w_1} & \cdots & \frac{\partial F_1}{\partial w_n} \\
\vdots & \ddots & \vdots \\
\frac{\partial F_m}{\partial w_1} & \cdots & \frac{\partial F_m}{\partial w_n}
\end{bmatrix}
$$

we divide each gradient by the corresponding learning rate $\alpha$ to obtain the original non-scaled gradient, and take its absolute value $\left|\frac{dE(y)}{dw}\right|$, since we are interested in the magnitude of the gradient and not its direction. $\Omega$ is the number of epochs to train for with full backpropagation at the beginning before starting partial backpropagation. As discussed earlier, we set $\Omega = 2$. For each layer $i = 1 : n$,

we create a mask $m_i$ that has the same shape of the gradient matrix $J$. We then set the corresponding mask value for each gradient in the layer $i$ to 0 if the gradient is smaller than the threshold $t$, and 1 otherwise. Thus, we have a list of masks for each layer: $m = (m_1, m_2, \ldots . m_n)$ where $n$ is the number of layers. We then perform weight update only on the weights corresponding to an entry of 1 in the corresponding mask $m_i$. For the rest of the batches in the third epoch and all the remaining epochs, we use the same mask $m$. Gradients and weight updates are only calculated for entries that correspond to 1 in the mask $m_i$. If the weight corresponds to an entry of 0 in the mask, its gradient is set to 0 and thus no weight update will happen to it (frozen weight). Therefore, the mask is a one-time calculation that depends on one batch in the third epoch, which makes it critical to have a reasonable batch size that contains all possible non-linear data combinations. In our experiments, we set the batch size to 100. Note that generating a mask requires one full iteration (one batch) of backpropagation. We have also experimented with changing the mask every several epochs (changing the mask each 35 epochs), and found out that it leads to the same performance as using the same mask for all epochs. Moreover, we experimented with performing full backpropagation at the last epoch in order to fine-tune the network so that the frozen weights could possibly change if needed. However, we found that the accuracy maintains as it is, implying that the frozen weights with least gradients are already optimized from the first two epochs, which validates our methodology.

## 4 EXPERIMENTS

### 4.1 RESULTS WITH CNNS

We start our experiments on the CIFAR-10 dataset using several CNN architectures. The CIFAR-10 dataset consists of 60000 32x32 colour images with 10 classes and 6000 images per class. There are 50,000 training images and 10,000 test images. Figure 3 shows the training and validation plots for both full backpropagation (FB) and experimental technique (PB). We use four different architectures: VGG19 (Simonyan & Zisserman, 2015), ResNet-110 (He et al., 2016), DenseNet-121 (Huang et al., 2017) and LeNet-5 (LeCun, 1998). The construction details are as follows: For VGG19, we construct the number of channels as [$2 \times 64$, 'M', $2 \times 128$, 'M', $4 \times 256$, 'M',$4 \times 512$, 'M', $4 \times 512$, 'M'] with $3 \times 3$ convolutions, where 'M' stands for max-pooling. For ResNet-110, we follow the proposed CIFAR-10 architecture in (He et al., 2016) (i.e. 54 blocks with [16, 32, 64] channels). For DenseNet-121, we use [6,12,24,16] blocks with a growth rate of 12. Finally, for LeNet-5, the number of channels are [8,32]. Table 1 demonstrates the results obtained, while Table 2 demonstrates the average performance over 3 runs.

The lowest freezing percentage is witnessed by residual networks. This is expected since these type of networks only learn the residual information and thus a lot of redundancy is already removed. Moreover, we find that VGG19 experiences 9.4% parameters with zero gradients by default (without applying our experimental technique), while LeNet-5 experiences 31.8% by default. Moreover, to demonstrate best performance of our methodology (i.e. highest freezing percentage with lowest accuracy drop), we slightly tweak the original threshold value obtained as discussed in section 3.1, and report the tweaked threshold values in Table 1.

It is worth noting that when training the networks, we avoid using any regularization techniques. In particular, we don't use data augmentation, neither weight decay or dropout. We focus on delivering our experiments under normal settings. Models are all trained for 110 epochs. We start with a learning rate of 0.001 and decay the learning rate by a factor of 0.1 at epoch 40 and 80. We train all networks with AMSGrad (Reddi et al., 2018) which takes the maximum between the current and the previous exponentially weighted average of gradients.

### 4.2 RESULTS WITH RNNS (USING LSTMS)

In order to experiment on complex real-life applications, we trained an image captioning model with visual attention mechanism as proposed in (Xu et al., 2015) on the Flickr8k dataset (Hodosh et al., 2013) using our experimental technique (PB). Flickr8k is a dataset that contains 8,000 images with up to 5 captions for each image. We divide the dataset into 6,000 training images and 1,000 validation and testing images, each. In our setup, we use the 5 captions for each image (if all 5 are present) and randomly sample from any present caption (5-present captions) times if the 5 captions are not present, resulting in 30,000 training captions and 5,000 validation and testing captions, each. We

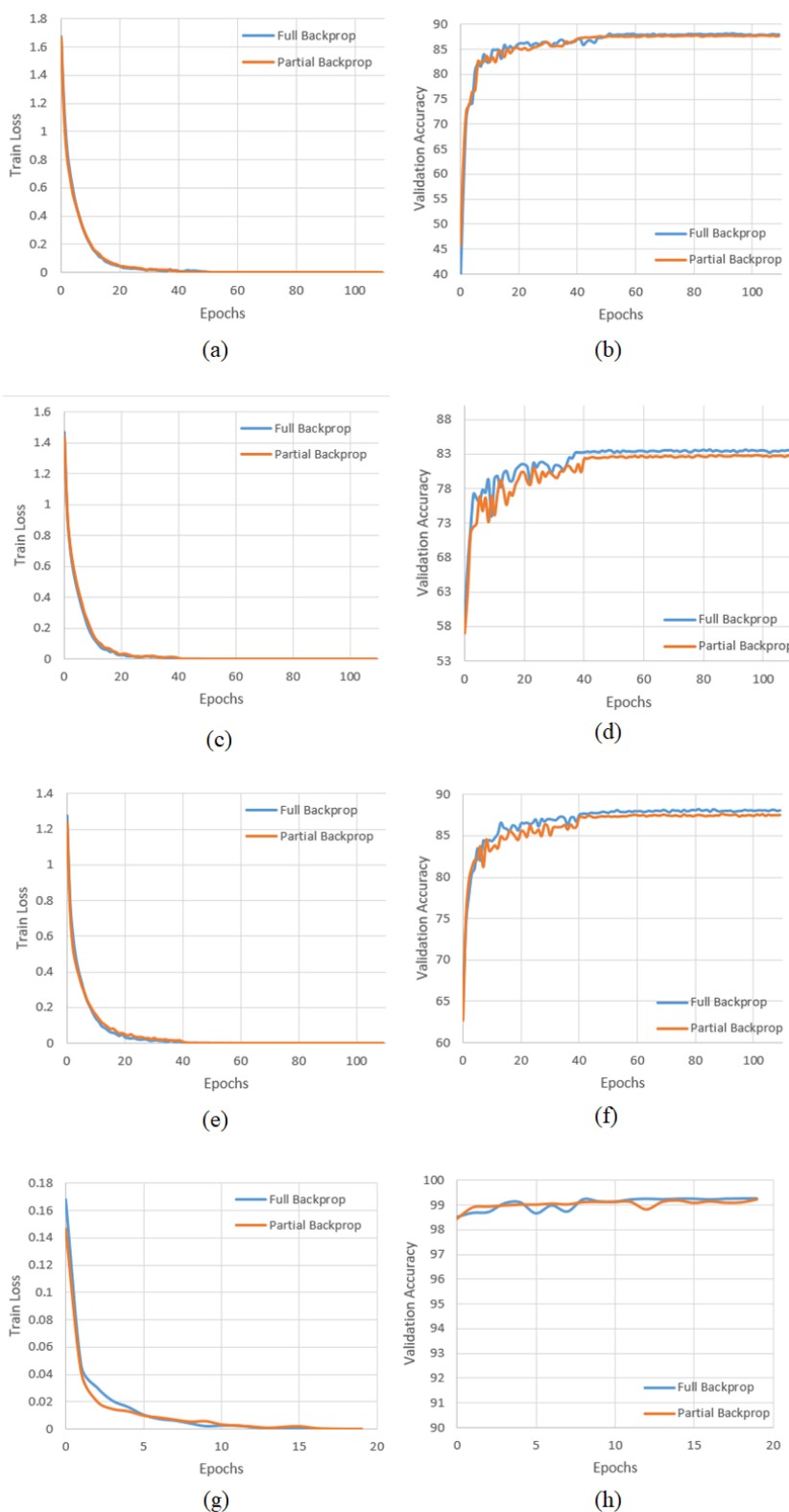

Figure 3: The training loss (left) and validation accuracy (right) plots for (a,b) VGG19 trained on CIFAR10, (c,d) ResNet-110 trained on CIFAR10, (e,f) DenseNet-121 trained on CIFAR10 and (g,h) LeNet-5 trained on MNIST

| Network | Dataset | Threshold | V. Accuracy (FB) | V. Accuracy (PB) | Frozen |
|---------|---------|-----------|------------------|------------------|--------|
| VGG19 | CIFAR10 | 0.04 | 88.12 % | 87.88% | 79.68% |
| ResNet-110 | CIFAR10 | 0.19 | 83.72% | 82.81% | 49.85% |
| DenseNet-121 | CIFAR10 | 0.45 | 88.20% | 87.63% | 69.07% |
| LeNet-5 | MNIST | 0.50 | 99.25% | 99.22% | 93.70% |

Table 1: Performance on CIFAR10 and MNIST datasets for 4 different types of CNNs with the threshold, validation accuracy for full backpropagation (FB), validation accuracy for experimental technique (PB) (PB) and the freezing ratio in parameters which is calculated as: (total gradients - non-zero gradients / total gradients) × 100

| Network | Dataset | Threshold | V. Accuracy (FB) | V. Accuracy (PB) | Frozen |
|---------|---------|-----------|------------------|------------------|--------|
| VGG19 | CIFAR10 | 0.04 | 88.12 % | 87.70% | 80.45% |
| ResNet-110 | CIFAR10 | 0.19 | 83.72% | 82.83% | 51.53% |
| DenseNet-121 | CIFAR10 | 0.45 | 88.20% | 87.52% | 69.33% |
| LeNet-5 | MNIST | 0.50 | 99.25% | 99.22% | 92.36% |

Table 2: Average Performance over 3 runs on CIFAR10 and MNIST datasets for 4 different types of CNNs with the threshold, validation accuracy for full backpropagation (FB), validation accuracy for our experimental technique (PB) and the freezing ratio in parameters which is calculated as: (total gradients - non-zero gradients / total gradients) × 100

resize all images to 256 × 256 pixels. We use a single layer LSTM with a hidden size of 512. The batch size is set 60. We use the Adam optimizer (Kingma & Ba, 2015) with an initial learning rate of 5e-4 and anneal the learning rate by a factor of 0.8 once the BLEU-4 score shows no improvement for 3 consecutive epochs. The word embedding and attention size is set to 512. We train for a maximum of 15 epochs with early stopping if the validation BLEU-4 score has not improved for 10 consecutive epochs. When sampling, we use a beam size of 3. We use BLEU (Papineni et al., 2001) with up to 4 grams (BLEU-1, BLEU-2, BLEU-3, BLEU-4) as our evaluation metric. We experiment on the soft attention variant of Xu et al. (2015). The best BLEU-4 score obtained when training with full backpropagation is 0.099. When applying our experimental technique (PB), using a threshold of 0.05, we obtain 61.81% redundant parameters which are frozen from the third epoch onwards. Under this setting, we obtain a higher BLEU-4 score of 0.101 than the fully trained model. The freezing ratio is calculated as: (total gradients - non-zero gradients / total gradients) × 100. Figure 4 shows some generated captions. For evaluation results on all BLEU scores, see Figure 5.

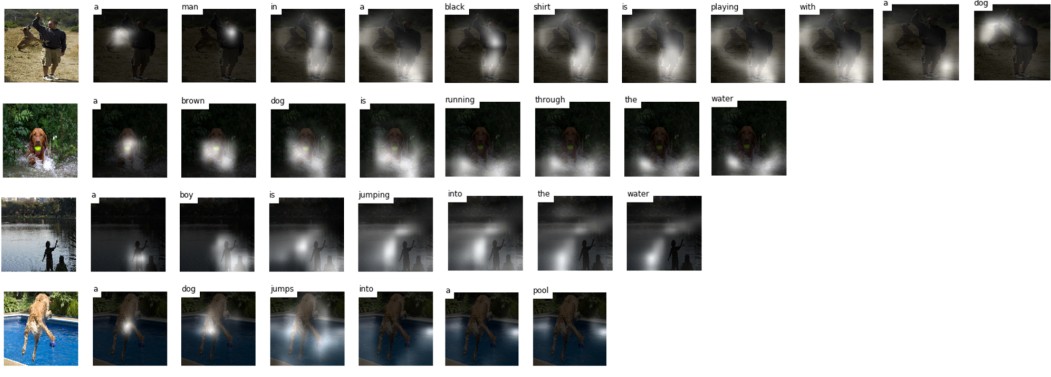

Figure 4: Results on training an image captioning model with visual attention mechanism (Xu et al., 2015) on the Flickr8k dataset using experimental technique (PB). The predicted word and attention weights for each timestep are shown for 4 generated captions of different lengths

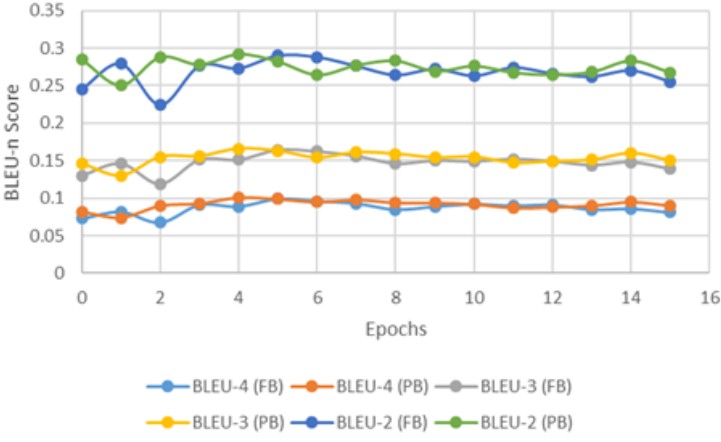

Figure 5: BLEU-n scores (where n is the number of grams) reported on the the Flickr8k dataset for image captioning with attention mechanism using both full backpropagation (FB) and our experimental technique (PB).

## 5 CONCLUSION

We provided an experimental study on the importance of a neural network weights, and to which extent do they need to be updated. Through our experiments, we emphasized the number of redundant parameters that carry no informative gradient, which if frozen from the third epoch onwards, slightly effect (and in sometimes do not) the overall accuracy of the model. To prove our concern, we ran experiments on the MNIST and CIFAR10 datasets using several CNN architectures (VGG19, ResNet-110 and DenseNet-121), as well as the Flick8k dataset using an image captioning architecture composed of LSTM networks with attention mechanism. Our experiments successfully prove the concern of this paper.

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

## A    APPENDIX

We provide our base code in this appendix. We use PyTorch to implement our experiments. We modified publicly available codes for the architectures we used.

```python
import torch
import torch.nn as nn
import torch.nn.functional as F
import torchvision.transforms as transforms
import torchvision.datasets as datasets
import matplotlib.pyplot as plt
import numpy as np
import math
from collections import import Counter
from tensorboard_logger import configure, log_value
import torch.backends.cudnn as cudnn
get_ipython().run_line_magic("matplotlib", "inline")

batch_size = 100
epochs = 110
# Set normal_train to True to disable partial backpropagation
normal_train = False
mnist = False
cifar10 = True
tensorboard = True
best_prec1 = 0
print_freq = 100
calculated = False
name = "densenet"
save = True
load_checkpoint = False
start_epoch = 0
# Epoch after which to start partial backprop (epoch starts at 0)
start_pb_epoch = 1
cudnn.benchmark = True

def zeroGrad(optimizer, net, masks):
    if masks is None:
        masks = list()
        for param in net.parameters():
            gradients = param.grad.detach()
            gradients = gradients / optimizer.param_groups[0]["lr"]
            all_positive = gradients.abs()
            thresh = 0.45
            mask = torch.ones_like(all_positive)
            mask[all_positive<=thresh] = 0
            param.grad*=mask
            masks.append(mask)
        return masks, net

    else:
        for i,param in enumerate(net.parameters()):
            param.grad*=masks[i]
        return masks, net

def countZeroGrads(net):
    zeros = 0
    for param in net.parameters():
        grad = param.grad.detach()
```

```python
        zeros += torch.sum((grad == 0).int()).item()
    return zeros

def countNonZeroGrads(net):
    non_zeros = 0
    for param in net.parameters():
        grad = param.grad.detach()
        non_zeros += torch.sum((grad != 0).int()).item()
    return non_zeros

def count_parameters(model):
    return sum(p.numel() for p in model.parameters() if p.requires_grad)

def set_learning_rate(optimizer, new_lr):
    for param_group in optimizer.param_groups:
        param_group['lr'] = new_lr

transformation = transforms.Compose([transforms.ToTensor(),
                                    transforms.Normalize((0.4914, 0.4822, 0.4465),
                                                        (0.2023, 0.1994, 0.2010))])

train_dataset = datasets.CIFAR10(root="./cifar10data",
                                 train=True,
                                 download=True,
                                 transform=transformation)

test_dataset = datasets.CIFAR10(root="./cifar10data",
                                train=False,
                                download=True,
                                transform=transformation)
if tensorboard:
    configure("runs/%s"%(name))

train_loader = torch.utils.data.DataLoader(dataset = train_dataset,
                                           batch_size = batch_size,
                                           shuffle = True)

val_loader = torch.utils.data.DataLoader(dataset = test_dataset,
                                         batch_size = batch_size,
                                         shuffle = False)

######### LeNet and VGG19 #############
# from models import *
# model = CNN()
# model = VGG(VGG19)

########### RESNETCIFAR ###########
# from resnetcifar import resnet

# model = resnet.resnet110()

########### DENSENET ############
from models import *
model = densenet_cifar()
CUDA = torch.cuda.is_available()

if CUDA:
```

```python
    model = model.cuda()

criterion = nn.CrossEntropyLoss()
optimizer = torch.optim.Adam(model.parameters(), lr=0.001, amsgrad=True)

if load_checkpoint:
    checkpoint = torch.load('best_model_epoch_1')
    start_epoch = checkpoint['epoch']
    best_prec1 = checkpoint['best_prec1']
    model.load_state_dict(checkpoint['state_dict'])

class AverageMeter(object):

    def __init__(self):
        self.reset()

    def reset(self):
        self.val = 0
        self.avg = 0
        self.sum = 0
        self.count = 0

    def update(self, val, n=1):
        self.val = val
        self.sum += val * n
        self.count += n
        self.avg = self.sum / self.count

def accuracy(output, target, topk=(1,)):
    """Computes the precision@k for the specified values of k"""
    maxk = max(topk)
    batch_size = target.size(0)
    _, pred = output.topk(maxk, 1, True, True)
    pred = pred.t()
    correct = pred.eq(target.view(1, -1).expand_as(pred))

    res = []
    for k in topk:
        correct_k = correct[:k].view(-1).float().sum(0)
        res.append(correct_k.mul_(100.0 / batch_size))
    return re

def train(train_loader, model, criterion, optimizer, epoch):
    global calculated, masks
    #global learning_rate
    losses = AverageMeter()
    top1 = AverageMeter()

    # switch to train mode
    model.train()

    for i, (input, target) in enumerate(train_loader):
        target = target.cuda(async=True)
        input = input.cuda()
        input_var = torch.autograd.Variable(input)
        target_var = torch.autograd.Variable(target)
        # compute output
        output = model(input_var)
        loss = criterion(output, target_var)
```

```python
            # measure accuracy and record loss
            prec1 = accuracy(output.data, target, topk=(1,))[0]
            losses.update(loss.data[0], input.size(0))
            top1.update(prec1[0], input.size(0))
            # compute gradient and do SGD step
            model.zero_grad()
            loss.backward()

            if not normal_train:

                if calculated is False and epoch>start_pb_epoch:
                    calculated = True
                    masks, model = zeroGrad(optimizer, model, masks = None)
                else:
                    if epoch>start_pb_epoch:
                        masks, model = zeroGrad(optimizer, model, masks)

            if i % print_freq== 0:
                # Make sure gradients are zeroed out
                print("Number of Zero Gradients: ", countZeroGrads(model))
                #Make sure optimizer parameters and model parameters are the same
#                 pm = set(p.data_ptr() for p in model.parameters())
#                 po = set(p.data_ptr() for p in optimizer.param_groups[0][params])
#                 print("Pointer Status:", pm == po)

            optimizer.step()

            if i % print_freq == 0:
                nonzeros_param = countNonZeroGrads(model)
                original_param = count_parameters(model)
                reduction = ((original_param - nonzeros_param) / original_param) * 100
                print("Epoch: [{0}][{1}/{2}]\t"
                        "Loss {loss.val:.4f} ({loss.avg:.4f})\t"
                        "Prec@1 {top1.val:.3f} ({top1.avg:.3f})\t".format(epoch,i,
                                                    len(train_loader)
                                                    loss=losses,
                                                    top1=top1))
                print("Non-Zero Gradients: {}".format(nonzeros_param)
                print("Total Gradients: {}".format(original_param)
                print("Reduction: {:.3f}%".format(reduction)

        # log to TensorBoard
        if tensorboard:
            log_value("train_loss", losses.avg, epoch)
            log_value("train_acc", top1.avg, epoch)

def validate(val_loader, model, criterion, epoch):
    losses = AverageMeter()
    top1 = AverageMeter()
    # switch to evaluate mode
    model.eval()
    for i, (input, target) in enumerate(val_loader):
        target = target.cuda(async=True)
        input = input.cuda()
        input_var = torch.autograd.Variable(input, volatile=True)
        target_var = torch.autograd.Variable(target, volatile=True)
        # compute output
        output = model(input_var)
```

```python
            loss = criterion(output, target_var)
            # measure accuracy and record loss
            prec1 = accuracy(output.data, target, topk=(1,))[0]
            losses.update(loss.data[0], input.size(0))
            top1.update(prec1[0], input.size(0))

            if i % print_freq == 0:
                print("Test: [{0}/{1}]\t"
                "Loss {loss.val:.4f} ({loss.avg:.4f})\t"
                "Prec@1 {top1.val:.3f} ({top1.avg:.3f})".format(i,
                                                len(val_loader),
                                                loss=losses,
                                                top1=top1))

        print("* Prec@1 {top1.avg:.3f}".format(top1=top1))
        # log to TensorBoard
        if tensorboard:
            log_value("val_loss", losses.avg, epoch)
            log_value("val_acc", top1.avg, epoch)
        return top1.avg

for epoch in range(start_epoch, epochs):
    if epoch == 40:
        set_learning_rate(optimizer, new_lr = 0.0001)
    if epoch == 80:
        set_learning_rate(optimizer, new_lr = 0.00001)

    train(train_loader, model, criterion, optimizer, epoch)
    prec1 = validate(val_loader, model, criterion, epoch)
    is_best = prec1 > best_prec1
    best_prec1 = max(prec1, best_prec1)
    if save and is_best:
        torch.save({"epoch": epoch,
                    "state_dict": model.state_dict(),
                    "best_prec1": best_prec1},
                   "best_model_epoch_{}_score_{}.pth".format(epoch, prec1))

print("Best accuracy: ", best_prec1.item())
```

