# OpenReview forum: "HOW IMPORTANT ARE NETWORK WEIGHTS? TO WHAT EXTENT DO THEY NEED AN UPDATE?"
_ICLR.cc/2020/Conference — Reject_

### Official Review · AnonReviewer1 · 2019-10-21
**Official Blind Review #1**

**Rating:** 1

**Review:**


In this paper, the authors performed an empirical study on the importance of neural network weights and to which extent they need to be updated. Some observations are obtained such as from the third epoch on, a large proportion of weights do not need to be updated and the performance of the network is not significantly affected.

Overall speaking, the qualitative result in the paper has already been discovered in many previous work, although the quantitative results seem to be new. However, there is large room to improve regarding the experimental design and the comprehensiveness of the experiments. Just name a few as follows:

1)	For different models and different tasks, the quantitative results are different. There is no deep discussion on the intrinsic reason for this, and what is the most important factor that influences the redundancy of weight updates. The authors came to the conclusion that from the third epoch on, no need to update most of the weights. “3” seems to be a magic number to me. Why is it? No solid experiments were done regarding this, and no convincing analysis was made.

2)	The datasets used in the experiments are not diverse enough and are not of large scale. For example, the CIFA-10 and MNIST datasets are relatively of small scale. What if the datasets are much larger like ImageNet. In such more complicated case, will the weight updates still be unnecessary? Will the ratio and the epoch number change? What is the underlying factor determining these? For another example, there are many NLP datasets for language understanding and machine translation, which are of large scale. Why choosing an image captioning dataset (which I do not agree to be real-life experiments when compared with language understanding and machine translation)? Can the observations generalizable to more complicated tasks and datasets?

3)	The models studied in the paper are also a little simple, especially for the text task. Why just using a single-layer LSTM? Why not popularly used Transformer?

As a summary, for an empirical study to be convincing, the tasks, datasets, scales, model structures, detailed settings, and discussions are the critical aspects. However, as explained above, this paper has not done a good job on these aspects. Significantly more work needs to be done in order to make it an impactful work.

*I read the author rebuttal, but would like to keep my rating unchanged.

**Experience Assessment:**

I have published in this field for several years.

**Review Assessment: Checking Correctness Of Derivations And Theory:**

N/A

**Review Assessment: Checking Correctness Of Experiments:**

I carefully checked the experiments.

**Review Assessment: Thoroughness In Paper Reading:**

I read the paper thoroughly.

---

> ### Author Response · Authors · 2019-11-10
> **response**
>
> Thanks for your feedback.
> For point number 1, we will try to further discuss and analyze this in a convincing manner.
>
> For point 2, we have chosen Image Captioning as it combines both vision (CNN operating on large-sized images) and language (RNNs for language modeling), and we believe that it reflects image understanding and language modeling at the same time, which is the reason why we chose it. As for using transformers, they themselves are very sensitive to train, and in our study we only focus on delivering our concern, without focusing on using "the best model".
>
> For point 3, For your comment on the model simplicity, we believe that the used models though they are simple (such as one layer LSTM) they are sufficient enough to proof the concept as this is mainly a conceptual paper to theoretically prove that it is possible to freeze certain insignificant weights in a neural network.

---

### Official Review · AnonReviewer2 · 2019-10-22
**Official Blind Review #2**

**Rating:** 3

**Review:**

The paper presents the empirical observation that one can freeze (stop updating) a significant fraction of neural network parameters after only training for a short amount of time, without hurting final performance too much. The technical contribution made by this paper is an algorithm for determining which weights to freeze, called partial backpropagation, and an empirical validation of the algorithm on various models for image recognition.

The observation that weights can be frozen is somewhat interesting, although similar findings have been reported before.
It's not clear the proposed algorithm is useful. The authors mention that fully parameterized models are expensive to run, but they don't demonstrate any speed-ups using their approach. Such speed-up would also not be expected since the forward pass of the algorithm cannot get faster by freezing weights, and the impact on the backward pass is limited. I'd be willing to raise my rating if the authors can convince me of the usefulness of their algorithm.

**Experience Assessment:**

I have published in this field for several years.

**Review Assessment: Checking Correctness Of Derivations And Theory:**

I assessed the sensibility of the derivations and theory.

**Review Assessment: Checking Correctness Of Experiments:**

I assessed the sensibility of the experiments.

**Review Assessment: Thoroughness In Paper Reading:**

I read the paper at least twice and used my best judgement in assessing the paper.

---

> ### Author Response · Authors · 2019-11-10
> **response**
>
> Thank you very much for your very useful feedback. This is mainly a conceptual paper to theoretically prove that it is possible to freeze certain insignificant weights in the neural network. By right if the number of the updates needed to be performed on the parameters is much lesser, the backward pass time is shorter, which will eventually speed up the whole process. However, freezing individual weights within a layer is not possible by any current deep learning framework (only freezing a complete layer will all its weights is possible), and developing the code from scratch to perform as efficiently as any framework would will take a considerable amount of time. Nevertheless, the purposes of this paper is to prove the concept theoretically with sufficient empirical evidences.

---

### Official Review · AnonReviewer3 · 2019-10-23
**Official Blind Review #3**

**Rating:** 1

**Review:**

This paper studies the importance of a neural networks weights and to which extend do they need to be updated. Particularly, the authors show that freezing weights which have small gradient in the very beginning of the training only results in a very slight drop in the final accuracy.

This paper should be rejected because (1) the paper only provides some empirical results on freezing network network weights, I don't think there are much insights and useful information; (2) To my knowledge, the phenomenon that only a few parameters are important has been observed before by many papers.

Given that, I vote for a rejection.

**Experience Assessment:**

I have published one or two papers in this area.

**Review Assessment: Checking Correctness Of Derivations And Theory:**

N/A

**Review Assessment: Checking Correctness Of Experiments:**

I assessed the sensibility of the experiments.

**Review Assessment: Thoroughness In Paper Reading:**

I read the paper at least twice and used my best judgement in assessing the paper.

---

> ### Author Response · Authors · 2019-11-10
> **response**
>
> Thank you for your feedback. To best of our knowledge, the previous studies have observed the process of freezing complete layers in a neural network unlike our studies that investigates the importance of the individual gradient parameters even in different layers without the need to freeze the complete layer and here lies the uniqueness of our study.

---

### Decision · Program_Chairs · 2019-12-19

**Decision:**

Reject

**Comment:**

The authors demonstrate that starting from the 3rd epoch, freezing a large fraction of the weights (based on gradient information), but not entire layers, results in slight drops in performance.

Given existing literature, the reviewers did not find this surprising, even though freezing only some of a layers weights has not been explicitly analyzed before. Although this is an interesting observation, the authors did not explain why this finding is important and it is unclear what the impact of such a finding will be. The authors are encouraged to expand on the implications of their finding and theoretical basis for it. Furthermore, reviewers raised concerns about the extensiveness of the empirical evaluation.

This paper falls below the bar for ICLR, so I recommend rejection.